# Satellite Retrieval of Aerosol Combined with Assimilated Forecast

Mayumi Yoshida[1], Keiya Yumimoto[2], Takashi M. Nagao[3], Taichu Tanaka[4], Maki Kikuchi[5], Hiroshi Murakami[5]

[1]Japan Aerospace Exploration Agency, Tsukuba, 305-8505, Japan, (Present affiliation is Remote Sensing Technology Center of Japan, Tsukuba, 305-8505, Japan)
[2]Research Institute for Applied Mechanics, Kyushu University, Fukuoka, 816-8580, Japan
[3]Atmosphere and Ocean Research Institute, Tokyo University, Chiba, 277-8564, Japan
[4]Meteorological Research Institute, Tsukuba, 305-0052, Japan
[5]Japan Aerospace Exploration Agency, Tsukuba, 305-8505, Japan

*Correspondence to*: Mayumi Yoshida (mayum@restec.or.jp)

**Abstract.** We developed a new aerosol satellite retrieval algorithm combining a numerical aerosol forecast. In the retrieval algorithm, the short-term forecast from an aerosol data assimilation system was used as a priori estimate instead of spatially and temporally constant values. This method was demonstrated using observation of the Advanced Himawari Imager onboard the Japan Meteorological Agency's geostationary satellite Himawari-8. Overall, the retrieval results incorporated strengths of the observation and the model, and complemented their respective weaknesses, showing spatially finer distributions than the model forecast and less noisy distributions than the original algorithm. We validated the new algorithm using ground observation data and found that the aerosol parameters detectable by satellite sensors were retrieved more accurately than a priori model forecast by adding satellite information. Further, the satellite retrieval accuracy was improved by introducing the model forecast instead of the constant a priori estimates. By using the assimilated forecast for a priori estimate, information from previous observations can be propagated to future retrievals, leading to better retrieval accuracy. Observational information from the satellite and aerosol transport by the model are incorporated cyclically to effectively estimate the optimum field of aerosol.

## 1 Introduction

Aerosols have a fundamental influence on the energy budget of the earth's climate system through the scattering and absorption of solar radiation. The fifth assessment report of the Intergovernmental Panel on Climate Change (IPCC 2014) stated that radiative forcing of the total aerosol effect in the atmosphere, including cloud adjustments due to aerosols, is –0.9 W m$^{-2}$. The report also highlighted that the range of uncertainties in these radiative forcing estimations remains large (−1.9 W m$^{-2}$ to −0.1 W m$^{-2}$). Identifying the frequency and properties of aerosols over the globe by satellite measurements is essential in estimating the radiation budget and the impacts of aerosols on climate systems.

In satellite aerosol remote sensing, not all aerosol properties can be accurately detected by satellite sensors, as there are more unknown aerosol parameters (e.g., particle size distributions, vertical density distribution, shape, refractive index) than the actual information obtained by the sensors. Most studies use assumptions or information about aerosol parameters, and limit

the number of parameters retrieved. For example, Higurashi and Nakajima (1999), and Fukuda et al. (2013) assumed fixed complex refractive indices ($1.5 - 0.005i$ in Higurashi and Nakajima (1999), and $1.503 - 7.16 \times 10^{-8}i$ for small mode particles and $1.445 - 1.00 \times 10^{-8}i$ for coarse mode particles based on sulfate and sea spray models, respectively, in Fukuda et al. (2013)), and retrieved the aerosol optical thickness and Ångström exponent. Some studies assumed aerosol particle models according to the location and season. For example, Kaufman et al. (1997), Remer et al. (2005), and Levy et al. (2007)

estimated aerosol optical thickness and fine mode fraction over a dark target using the Moderate Resolution Imaging Spectroradiometer (MODIS), by selecting the fine-dominate particle models as a function of geography and season. Hsu et al. (2004) retrieved the optical thickness and type of aerosols over desert regions using blue channels (< 500 nm), where the surface reflectance was relatively low, assuming dust or a mixture of dust and smoke depending on the region and season. Jeong et al. (2016) used a priori information according to the location and season. However, these studies did not take

temporal changes into account. Because it is impossible to completely fix aerosol type as a function of geographical location and season, the unrealistic assumptions hence implemented lead to one of the major causes of retrieval error.

Aerosol data assimilation methods using satellite data have also been developed to obtain better initial conditions for the aerosol transport model. The aerosol data assimilation study was first developed with Low Earth Orbit (LEO) satellites

(Benedetti et al., 2009; Saide et al., 2013; Dai et al., 2014; Rubin et al., 2015; Yumimoto et al., 2015). In recent years, assimilation studies have been extended to using geostationary satellites with large spatial coverage and fine observation frequencies (Saide et al., 2014; Lee et al., 2016; Yumimoto et al., 2016; Yumimoto et al., 2018; Die et al., 2019; Jin et al., 2019).

Due to the development of such assimilation studies, the satellite data have contributed to improving aerosol forecast simulations. However, no studies have utilized assimilated model forecast as a priori estimate of the retrieval. Since satellite sensors cannot accurately detect all parameters, and unrealistic assumptions of aerosol parameters are a major cause of retrieval errors as mentioned above, adding the model information is expected to improve the retrieval accuracy. Therefore, in this study, we utilize the forecast of an aerosol transport model for a priori estimates of the retrieval. This allows the

aerosol information in the aerosol transport model to be used for retrieval. By using the assimilated forecast, information from previous satellite observations can be propagated to future satellite retrievals through the aerosol transport model.

The sections in this study are organized as follows: Section 2 explains the retrieval methodology in detail. Section 3.1 presents the results of application to the Advanced Himawari Imager (AHI) onboard Himawari-8. Section 3.2 describes the

validation of the estimations using ground observations, and Section 3.3 tests the worst-case scenario. Finally, Section 4 summarizes our findings.

## 2 Methodology

The aerosol retrieval algorithm in this study is based on Yoshida et al. (2018). As a priori estimate of the retrieval, the algorithm introduces aerosol forecast from a transport model that has assimilated previous satellite observations. Given the
general applicability of the retrieval algorithm by Yoshida et al. (2018), the methodology explained in this section can also be applied to various sensors. Here, we demonstrate the algorithm using the Himawari-8/AHI whose assimilation system is operationally available. The AHI has six observation bands from visible to near-infrared wavelength ranges, and observes the top of atmosphere radiance at a resolution of 0.5-2.0 km over Asia and Oceania at 10-minute intervals (Bessho et al., 2016).


Figure 1 depicts an overview of the algorithm, showing the process of using forecast data for a priori estimates of the retrieval. In the original retrieval process, the Level-2 (L2) aerosol optical thickness at 500 nm ($\tau$), Ångström exponent between 400 and 600 nm ($\alpha$), and single-scattering albedo at 500 nm ($\omega$) are retrieved using Level-1 (L1) AHI-observed radiance every 10 minutes around time T0 as per Yoshida et al. (2018). The Level-3 (L3) $\tau$ and $\alpha$ at T0 are then estimated
using L2 products in one hour by an hourly-combined algorithm (Kikuchi et al., 2018). The hourly-combined algorithm is a method that (1) minimizes cloud contamination using the difference between aerosol and cloud spatiotemporal variability characteristics, and (2) interpolates the aerosol retrievals using one hour of data and the movement of clouds within the hour (see Kikuchi et al., 2018 for more details).

The L3 $\tau$ at T0 is then assimilated into a global aerosol transport model by the 2D-Var assimilation system (Yumimoto et al., 2018). For the aerosol transport model, we use MASINGAR (Model of Aerosol Species IN the Global AtmospheRe; Tanaka et al., 2003; Tanaka and Chiba, 2005) developed at the Meteorological Research Institute (MRI) of the Japan Meteorological Agency (JMA). MASINGAR covers the major tropospheric aerosol components (i.e., black and organic carbon, mineral dust (10-size bins), sea salt (10-size bins), sulfate aerosols)) and their precursors (e.g., sulfur dioxide ($SO_2$), dimethyl sulfide,
terpenes)), and is coupled online with an atmospheric general circulation model (MRI-AGCM3; Yukimoto et al., 2012). The model's grid resolution is set to horizontal Gaussian TL479 (960 x 480 grids, about 0.375 degree) and 40 vertical layers in hybrid sigma-pressure coordinates from the ground to 0.4 hPa. The integration time step is set to 600 seconds. Anthropogenic emissions of $SO_2$, black and organic carbon are taken from the MACCity emission inventory (Granier et al., 2011). Daily biomass burning emission flux is taken from the Global Fire Assimilation System (GFAS, Kaiser et al., 2012)
version 1.2 provided by the European Centre of Medium Range Forecast (ECMWF). The horizontal wind components and sea surface temperature are nudged toward the global analyses and forecasts of JMA (GANAL). The forecast from the

assimilation system serves as the operational sand and dust forecasting by JMA, the aerosol property model product in the JAXA Himawari Monitor (https://www.eorc.jaxa.jp/ptree/index.html), and a member of the ICAP multi-model ensemble (MME) (Xian et al., 2019). The volume concentration (then $\tau$) of each aerosol component at the next time (T1) is then forecasted using the assimilated aerosol transport model.

In the new retrieval process, we retrieve the L2 aerosol properties ($\tau$, $\alpha$, and $\omega$) from AHI-observed radiance at T1 using these L4 forecasts for a priori estimates of the retrieval. In this way, the information from previous observations at T0 is used for the next aerosol retrievals at T1 through the aerosol transport model. The retrieval obtained at T1 is further used in the same way to derive the retrieval at the following time step (T=T2, not shown) by using L4 forecasts for a priori estimate. Figure 6 compares the improved retrieval with the original retrieval at T1 as later described in Section 3.2. The methodology for using the forecast as a priori estimates of the retrieval is detailed as follows: In the retrieval process, the final retrieval parameters ($\tau$, $\alpha$, and $\omega$) are calculated from the set of aerosol parameters ($\tau$, external mixing ratio of dry volume concentration of fine particles $\eta_f$, and external mixing ratio of the dry volume concentration of dust particles for the coarse model $\eta_c^{dst}$) defined by Yoshida et al. (2018). Here, the imaginary part of the refractive index ($m_i$) for the fine aerosol model varies with change in $\eta_c^{dst}$ such that the fine and coarse models exhibit the same $\omega$ at 500 nm (see Yoshida et al., 2018 for more details). The $\alpha$ and $\omega$ are calculated from the retrieved $\eta_f$ and $\eta_c^{dst}$ (i.e., $m_i$ for fine aerosol model) using the tables previously calculated by radiative transfer code called the System for the Transfer of Atmospheric Radiation, whose development was initially led by the University of Tokyo (STAR, Nakajima and Tanaka 1986, 1988; Stamnes et al., 1988). The detailed aerosol setting is explained in Yoshida et al. (2018), and is outlined in Appendix A. Appendix B shows the relationship of the final retrieval parameters ($\alpha$ and $\omega$) with the set of aerosol parameters ($\eta_f$ and $\eta_c^{dst}$). We retrieve the aerosol parameters ($\tau$, $\eta_f$, and $\eta_c^{dst}$) using an optimal estimation method (Rodgers 2000). The state vector of a set of aerosol parameters $\boldsymbol{x} = \{\tau, \eta_f, \eta_c^{dst}\}$ is derived by minimizing object function $J$ (Eq. (1)). It uses the measurement vector of a gas-corrected observed reflectance set $\boldsymbol{R} = \{\rho_i^{obs'}, i = 1, ..., N\}$ and simulated TOA reflectance $\boldsymbol{F}(\boldsymbol{x})= \{\rho_i^{sim}, i = 1, ..., N\}$, where N is the channel number.

$$J = [\boldsymbol{R} - \boldsymbol{F}(\boldsymbol{x})]^T \boldsymbol{S}_e^{-1}[\boldsymbol{R} - \boldsymbol{F}(\boldsymbol{x})] + [\boldsymbol{x} - \boldsymbol{x}_a]^T \boldsymbol{S}_a^{-1}[\boldsymbol{x} - \boldsymbol{x}_a] \qquad (1)$$

where $\boldsymbol{x}_a = \{\tau_a, \eta_{f_a}, \eta_{c}^{dst}{}_a\}$ is the vector of a prior estimate of $\boldsymbol{x}$, and $\boldsymbol{S}_e$ and $\boldsymbol{S}_a$ are the covariance matrices of $\boldsymbol{R}$ and $\boldsymbol{x}_a$, respectively. The calculations of $\boldsymbol{R}, \boldsymbol{F}(\boldsymbol{x})$, and $\boldsymbol{S}_e$ are the same as those of Yoshida et al. (2018), but we apply canonical correlation analysis to find the optimal coordinate system, and converted $\boldsymbol{R}, \boldsymbol{F}(\boldsymbol{x})$, and $\boldsymbol{S}_e$ to the coordinate system whose dimension is reduced to the number of retrieved parameters (i.e., three). In the original retrieval process, we used spatially and temporally constant values of $\boldsymbol{x}_a$, and $\boldsymbol{S}_a$ that were derived from climatology analysis, and assumed that the non-diagonal component of covariance matrices was set to 0 (Yoshida et al., 2018).

To introduce more realistic a prior estimate and covariances into the retrieval process, we employ the forecast from the aerosol assimilation system instead of the constants. The model forecast includes the total aerosol optical thickness at 500 nm and 870 nm, and the absorption aerosol optical thickness at 500 nm derived from the modeled volume concentration and extinction cross section of each aerosol component (Yumimoto et al., 2017). We assign a priori estimate $x_a$ as follows: The model's total aerosol optical thickness at 500 nm is used for $\tau_a$. $\eta_{f_a}$ is derived from the ratio of total aerosol optical thickness between 500 nm and 870 nm. As the selection of $\eta_c^{dst}{}_a$, we use the model's $\omega$ as calculated from the total and absorption aerosol optical thickness at 500 nm.

The assimilation system uses an ensemble method to calculate the background error covariance matrix (Yumimoto et al., 2018). In the method, the ensemble was collected from forecast values within ±2 hours of the targeted hour of the five previous forecasts (Fig. 2). We employ this method to define $S_a$. The model ensemble enables $S_a$ to include the non-diagonal component and express the error of aerosol transport. However, $S_a$ from model ensemble may become too small when the model does not predict the aerosol event itself. For that reason, in order to estimate total $S_a$, we add a model absolute error ($S_a^A$) to the error estimated from the ensemble ($S_a^E$) as follows:

$$S_a = S_a^E + S_a^A = \begin{bmatrix} \sigma_{\tau_a}{}^2 & \sigma_{\tau_a \eta_{f_a}} & \sigma_{\tau_a \eta_c^{dst}{}_a} \\ \sigma_{\tau_a \eta_{f_a}} & \sigma_{\eta_{f_a}}{}^2 & \sigma_{\eta_{f_a} \eta_c^{dst}{}_a} \\ \sigma_{\tau_a \eta_c^{dst}{}_a} & \sigma_{\eta_{f_a} \eta_c^{dst}{}_a} & \sigma_{\eta_c^{dst}{}_a}{}^2 \end{bmatrix}, \tag{2}$$

$$\sigma_{\tau_a} = \sigma_{\tau_a}^E + \sigma_{\tau_a}^A, \tag{3}$$

$$\sigma_{\eta_{f_a}} = \sigma_{\eta_{f_a}}^E + \sigma_{\eta_{f_a}}^A, \tag{4}$$

$$\sigma_{\eta_c^{dst}{}_a} = \sigma_{\eta_c^{dst}{}_a}^E + \sigma_{\eta_c^{dst}{}_a}^A, \tag{5}$$

where $\sigma_{\tau_a}^E, \sigma_{\eta_{f_a}}^E$, and $\sigma_{\eta_c^{dst}{}_a}^E$ are the standard deviations of $\tau_a, \eta_{f_a}$, and $\eta_c^{dst}{}_a$, respectively, estimated from the ensemble. $\sigma_{\tau_a}^A, \sigma_{\eta_{f_a}}^A$, and $\sigma_{\eta_c^{dst}{}_a}^A$ are the same as those of the model absolute error.

$\sigma_{\tau_a}^A$ is set to $\sigma_{\tau_a}^G$ or $\sigma_{\tau_a}^M$ (whichever is larger) as follows:

$$\sigma_{\tau_a}^A = \begin{cases} \sigma_{\tau_a}^G & if \ \sigma_{\tau_a}^G \geq \sigma_{\tau_a}^M \\ \sigma_{\tau_a}^M & else \end{cases}, \tag{6}$$

where $\sigma_{\tau_a}^G$ is the Root Mean Square Error (RMSE) of the model's $\tau$ from ground observation (0.399 in Fig. 6 (c)), and $\sigma_{\tau_a}^M$ is the standard deviation of $\tau$ for five years as calculated by the free run model without assimilation. The free run model's spatial resolution is around 1.2 degrees, and the standard deviation is calculated for MAM (March, April and May), JJA (June, July and August), SON (September, October and November), and DJF (December, January and February) (Fig. 3). $\sigma_{\eta_{f_a}}^A$ (0.093) is calculated from RMSE of the model's α (0.223 in Fig. 6 (f)) at α of 1.2 and $\eta_c^{dst}$ of 0.5. $\sigma_{\eta_c^{dst}{}_a}^A$ is set to 0.5 because $\eta_c^{dst}$ takes a value from 0. to 1, and $\omega$ (which is uniquely determined by $\eta_c^{dst}$) has little correlation with the ground

observation data (Fig. 6 (i)). As the non-diagonal component of $S_a^A$ cannot currently be calculated from our limited database, we use the non-diagonal components of $S_a^E$ as those of $S_a$.

## 3 Results and Discussion

### 3.1 Results of application to Himawari-8

We applied the methodology described in Section 2 to the Himawari-8/AHI. We retrieved $\tau$, $\eta_f$ and $\eta_c^{dst}$, and then derived ω and α at 10-minute intervals from the calibrated L1 data subsampled at 0.05° using the method described in Section 2. The channels used for the retrieval are the same as those used by Yoshida et al. (2018), which are channels 1 (0.46 μm), 2 (0.51 μm), 3 (0.64 μm), 4 (0.86 μm), and 5 (1.6 μm) over land, and channels 4 and 5 over the ocean. As the number of satellite channels (two) used over the ocean is less than the number of retrieval parameters (three), not all parameters are stably retrieved by satellite data. Therefore, $\eta_c^{dst}$ over the ocean, which is the least sensitive to satellite observation, is set to 0 (i.e., non-absorbing aerosol) at this time, because the aerosol over the ocean is generally less absorbing than that over land, and using the model's $\eta_c^{dst}{}_a$ as $\eta_c^{dst}$ over the ocean leads to a worse estimation of $\tau$ (not shown). However, using non-absorbing aerosol over ocean causes a big problem in case of dust/smoke transported over the ocean, so we will use the model's $\eta_c^{dst}{}_a$ as $\eta_c^{dst}$ over the ocean after obtaining a better model's $\eta_c^{dst}{}_a$ in the future. Note that $\eta_c^{dst}$ over land is properly retrieved from satellite data (i.e., not set to 0) using the model's $\eta_c^{dst}{}_a$ as a priori estimate, since the number of satellite channels (five) used over land is greater than the number of retrieval parameters (three).

Figures 4 and 5 compare the retrieval results from the new algorithm using $S_a^E + S_a^A$ for $S_a$ with the original algorithm (Figs. 4/5 (a) and (b)). The retrieval results from the new algorithms using only $S_a^E$ for $S_a$ are also shown to evaluate the effect of $S_a^A$ on the retrieval result (Figs. 4/5 (c)). Figure 4 depicts the retrieved $\tau, \eta_f$ and $\eta_c^{dst}$ at 0200 UTC on May 19, 2016 when aerosols originating from wildfires near Lake Baikal in Russia reached Japan. The model's $x_a$ used for retrieval in Fig. 4 (b) and (c) is indicated in Fig. 4 (d). The $\sigma_{\tau_a}, \sigma_{\eta_{f_a}}$, and $\sigma_{\eta_c^{dst}{}_a}$ used for retrieval in Fig. 4 (b) are shown in Fig. 4 (e). The white regions indicate the area where retrieval is not executed due to the presence of clouds, etc. The two-hour forecasts starting from 0000 UTC on May 19 were assimilated with L3 merged $\tau$ at 0300, 0600, and 0900 UTC on May 18, and at 0000 UTC on May 19, and then used for a priori estimate (Fig. 4 (d)). Figure 5 is the same as Fig. 4 except for another case at 0500 UTC on May 7, 2017, when Asian dust was observed in Japan. The five-hour forecast starting from 0000 UTC on May 7 (and assimilated at 0300, 0600, and 0900 UTC on May 6, and at 0000 UTC on May 7) is used for a priori estimate (Fig. 5 (d)). These short-term forecasts with data assimilation are considered relatively realistic compared to long-term forecasts or a free run without assimilation (Yumimoto et al., 2018). If only model ensemble error is used for $S_a$ (Figs. 4 (c), 5 (c)), that is, the absolute error is not included in $S_a$, all retrieved parameters (especially $\eta_f$ and $\eta_c^{dst}$ over land) are highly dependent on a

priori estimate (Figs. 4 (d), 5 (d)). However, when using an appropriate $\boldsymbol{S_a}$ containing absolute error, the retrieved $\tau$, $\eta_f$, and $\eta_c^{dst}$ are updated by satellite data or remain close to a priori estimate depending on the location (Figs. 4 (b), 5 (b)). Specifically, spatially finer $\tau$ distributions than the model forecast are retrieved for cases of both wildfire aerosol (Fig. 4 (b))

and Asian dust (Fig. 5 (b)) due to the relatively coarser model horizontal resolution. Similar $\eta_f$ is retrieved over open ocean in Fig. 4 (b) and Fig. 5 (b), and the large $\eta_f$ (i.e., small particle) and small $\eta_f$ (i.e., large particle) are successfully retrieved in areas corresponding to wildfire aerosol (Fig. 4 (b)) and Asian dust (Fig. 5 (b)), respectively. This distribution is also expressed in the forecast model in Fig. 5 (d), but cannot be expressed sufficiently in the forecast model in Fig. 4 (d) because information about the aerosol particle size (e.g., $\alpha$, $\eta_f$) is not assimilated into the model. That is, by using an appropriate $\boldsymbol{S_a}$,

both the model and satellite data are used for estimating the aerosol parameters. In addition, the local noise in $\tau$ and $\eta_f$ is apparently reduced for this algorithm (Figs. 4 (b), 5 (b)) as compared with the original algorithm (Figs. 4 (a), 5 (a)). This will be discussed in Subsection 3.2.

## 3.2 Validation

We conducted a preliminary validation of our method by comparing the retrieved $\tau$, $\alpha$, and $\omega$ from the Himawari-8/AHI

with those from ground observation known as the Aerosol Robotic Network (AERONET). AERONET's $\tau$ and $\alpha$ were derived from Level 2.0 quality-assured Version 3 direct sun algorithm data (Giles et al., 2019; O'Neill et al., 2003), and $\omega$ was derived from Version 3 AERONET inversion products (Dubovik and King, 2000a; Dubovik et al., 2000b; Dubovik et al., 2002a; Dubovik et al., 2002b; Dubovik et al., 2006; Sinyuk et al., 2007). AERNET's $\omega$ at 500 nm was calculated from linear interpolation of $\omega$ at 440 nm and 675 nm. In this study, the 60 AERONET sites on the full disk of Himawari-8 were used for

the validation. We used the AERONET data averaged over 10 minutes of AHI observation time. For our retrieval data, we used $\tau$, $\alpha$, and $\omega$ estimated from AHI L1 radiance data subsampled at 0.05° nearest to the AERONET sites. Initial validation was conducted for six months (March, April, May, June, July, 2018, and February 2019). Long-term validation will be required in future studies.

Figure 6 compares the $\tau$, $\alpha$, and $\omega$ retrieved from the AHI with those from AERONET. The validation of $\alpha$ and $\omega$ is limited to cases where the simultaneously retrieved $\tau$ are greater than 0.3 because there is little information of $\alpha$ and $\omega$ from satellite observation for thin aerosol layer. The total number of validation points (14711, 14031, and 521) from this algorithm (Fig. 6 (b), (e), and (h)) is about 6-7% higher than those (13714, 13137, and 493) from the original algorithm (Fig. 6 (a), (d), and (g)), which means that the new algorithm successfully retrieved the aerosol in more cases than the original algorithm. Here,

the total number of validation points for $\omega$ is less than those for $\tau$ and $\alpha$, because the number of $\omega$ data from AERONET inversion products is less than those of $\tau$ and $\alpha$ from the direct sun measurements.

For the $\tau$ estimations (Fig. 6 (a), (b), (c)), the root mean square error (RMSE), mean bias (MB), and correlation (0.290, -0.099, and 0.758) from this algorithm (Fig. 6 (b)) are all better than those (0.399, -0.224, and 0.572) from the model forecast

(i.e., a priori estimate) in Fig. 6 (c), which means that satellite information is very effective for the retrieval of $\tau$. In addition, the RMSE (0.290) in Fig. 6 (b) is better than that (0.307) without the forecast model (Fig. 6 (a)), which means that the model information is also effective and the improved algorithm shows better performance than the original algorithm. The MB (-0.099) in Fig. 6 (b) is worse than that (-0.023) in Fig. 6 (a), probably because the large outlier in Fig. 6 (a) is improved in Fig. 6 (b). Figure 7 shows an example of the retrieval results of the outlier (red asterisks in Fig. 6 (a), (b), and (c)), and depicts

that the outlier of the original algorithm (Fig. 7(a)) is improved in the new algorithm by constraining $\tau$ to the model's $\tau_a$. In addition, the retrieval results around the red circles show that the new algorithm successfully retrieved the $\sigma_{\tau_a}$, $\sigma_{\eta_{f_a}}$, $\sigma_{\eta_c^{dst}}{}_a$ even where the original algorithm failed to retrieve. Thus, integrating the model and satellite information resulted in an improvement of the $\tau$ estimations.

For the $\alpha$ estimations (Fig. 6 (d), (e), (f)), large variance in the original method is considerably reduced by this method. The RMSE and correlation (0.271 and 0.581) from this algorithm (Fig. 6 (e)) are much better than those (0.429 and 0.353) from the original algorithm without the forecast model (Fig. 6 (d)), which indicates that the new algorithm could improve the precision of $\alpha$ estimations by adding more accurate $\alpha$ (RMSE of 0.223) information from the model. In addition, the MB (-0.052) from this algorithm (Fig. 6 (e)) is better than that (-0.057) from the model forecast (Fig. 6 (f)), due to the

improvement of negative bias in the large $\alpha$ in the model forecast. Thus, the new $\alpha$ can be retrieved with good accuracy by utilizing the relatively accurate model's $\alpha$ and correcting the bias by using the satellite data.

For the $\omega$ estimations (Fig. 6 (g), (h), (i)), the RMSE, MB, and correlation (0.035, -0.000, and 0.550) from this algorithm (Fig. 6 (h)) are better than those (0.048, -0.002, and 0.176) from the model forecast (Fig. 6 (i)), which indicates the

effectiveness of satellite information for $\omega$ retrieval. In addition, this algorithm improved RMSE, MB, and correlation by introducing the model forecast. Note that the current system assimilates only total $\tau$ (total amount of aerosols), and information about the fraction of fine and absorbing aerosols from the retrieval is not fed back to the model forecast. The improved retrieval accuracy of $\omega$ can be expected if the model's $\omega$ becomes more realistic in the future, such as by assimilating the satellite's $\omega$ to the model. Considering the validation results of $\tau$, $\alpha$, and $\omega$, this new algorithm effectively

improved the retrieval accuracy using information from both the model and the satellite by setting appropriate $\boldsymbol{S_a}$ and $\boldsymbol{S_e}$.

We also investigated the cause of the possible large deviation between the retrieved parameters from the new algorithm and the ground observation. Figures 8, 9, and 10 show the validation results of $\tau$, $\alpha$, and $\omega$, respectively, when the chi-square value ($\chi^2$) and the uncertainties of the retrieved three parameters ($\tau$, $\eta_f$, and $\eta_c^{dst}$) are smaller than a threshold. The chi-

square value ($\chi^2$) is calculated as follows:

$$\chi^2[R - F(x)] = [R - F(x)]^T S_e^{-1}[R - F(x)]/N, \quad (7)$$

and shows the closeness of the retrieved value to the observed value. The covariance matrix of the uncertainties of the retrieved parameters $S_{\hat{x}}$ is calculated using the law of error propagation, as follows:

$$S_{\hat{x}} = (A^T S_e^{-1} A)^{-1}, \quad (8)$$

where $A$ is the Jacobian matrix. $S_e$ is the covariance matrix of $R$, and calculated from sum of sensor noise and the uncertainty in TOA reflectance that results from surface reflectance uncertainty (Yoshida et al., 2018). In reality, the $S_e$ is almost determined by the uncertainty in TOA reflectance that results from surface reflectance uncertainty, because sensor noise is much smaller. Therefore, the $S_{\hat{x}}$ is mostly caused by the surface reflectance uncertainty. Figure 8 shows that RMSE for $\tau$ decreases as the threshold of $\chi^2$ or $S_{\hat{x}}$ becomes strict (i.e., decreases). On the other hand, RMSE for $\alpha$ (in Fig. 9) is not

dependent on the threshold of $S_{\hat{x}}$, but decreases as the $\chi^2$ threshold decreases. RMSE for $\omega$ (in Fig. 10) is little dependent on the threshold of $S_{\hat{x}}$ and $\chi^2$. Next, in Fig. 11 we investigated how the retrieved accuracy (difference between aerosol parameters retrieved from AHI and those of AERONET) depends on the model's (i.e., a priori) accuracy. The retrieved accuracy of $\alpha$ and $\omega$ has strong linear relationships (a correlation of 0.801, and 0.739, respectively) to the model's accuracy, while that of $\tau$ has a moderate linear relationship (a correlation of 0.622). Summarizing these results, the retrieved accuracy

of $\tau$ depends on all of the closeness to the observed value, accuracy of the surface reflectance estimation, and accuracy of a priori estimate, while the accuracy of a priori estimate is critical for the retrieved accuracy of $\alpha$ and $\omega$. Thus, introducing more realistic a priori estimates in the new retrieval algorithm instead of the constant values in the original algorithm led to the improvement of RMSE. It is also shown that the improvement of a numerical aerosol forecast by improving the aerosol transport model and the assimilation method, and increasing the assimilation frequency may further improve the retrieval

accuracy in the future.

**3.3 Worst-case scenario**

We have shown that the new retrieval algorithm using the forecast of an aerosol transport model improves the retrieval accuracy. However, in order to use this algorithm constantly (such as in an operational system), the effects of the model forecast (a priori estimate) that deviate from reality must be examined, because the model forecast may miss an aerosol event.

Therefore, we conducted a sensitivity test to investigate the impact on the retrieval results of using unrealistic forecast as a priori estimate. Figure 12 shows the retrieval results on the same day as in Fig. 4, except for using the forecast on another day (April 27, 2018) as a priori estimate of the retrieval (Fig. 12 (d)). If only $S_a^E$ is used as $S_a$ (Fig. 12 (c)), all parameters (especially $\eta_f$ and $\eta_c^{dst}$) are retrieved unrealistically by being dependent on the unrealistic a priori estimate. However, when using an appropriate $S_a$ (Eq. (2)), the retrieved parameters are well-updated by satellite data with less dependence on

unrealistic a priori estimate (Fig. 12 (b)). Even in such an extremely worst-case scenario, this new algorithm is apparently not significantly worse than the current algorithm, especially where the model forecast is missing an aerosol event, which may occur in the model forecast for natural aerosols (e.g., mineral dust and smoke from biomass burning).

## 4 Summary

We developed a new satellite aerosol retrieval algorithm combining a numerical aerosol forecast. In the retrieval algorithm, the short-term forecast from an aerosol data assimilation system was used for a priori estimate instead of spatially and temporally constant values. This is the first study that utilizes the assimilated model forecast of aerosol as a priori estimate of the satellite retrieval. We applied this new algorithm to the Himawari-8/AHI and confirmed that the aerosol parameters detectable by satellite sensors were retrieved more accurately (RMSE of 0.290 for $\tau$ and 0.035 for $\omega$) than a priori model forecast (RMSE of 0.399 for $\tau$ and 0.048 for $\omega$) by adding satellite information. Moreover, the satellite retrieval accuracy was improved (RMSE of 0.290 for $\tau$, 0.271 for $\alpha$, and 0.035 for $\omega$) by using the model forecast as compared with those using constant a priori estimates (RMSE of 0.307 for $\tau$ and 0.429 for $\alpha$, and 0.039 for $\omega$). As a result, aerosol retrievals were improved by effectively incorporating both model and satellite information, depending on each covariance. By using the assimilated forecast as a priori estimate, information from previous observations can be propagated to future retrievals, thereby leading to better retrieval accuracy. In this way, satellite observation and model simulation are used synergistically to continuously estimate the optimum field of aerosol. Future work would include applying the methodology proposed in this study to polar-orbiting satellites and combining them with geostationary satellite measurements, in order to offer consistent geostationary and polar-orbiting estimates, and thereby improve aerosol properties over the globe.

## Appendix A: Aerosol Setting

We assume that the aerosol model is an external mixture of fine and coarse particles ($\eta_f$ is the external mixing ratio of the dry volume concentration of fine particles). We set the fine aerosol model based on the average properties of fine mode for categories 1–6 by Omar et al. (2005). For the coarse aerosol model, we set the external mixture of the pure marine aerosol on the basis of the model illustrated by Sayer et al. (2012) and the dust model based on the coarse model of category 1 (dust) as illustrated by Omar et al. (2005). $\eta_c^{dst}$ is the external mixing ratio of the dry volume concentration of dust particles for the coarse model.

Regarding each aerosol size, we use a monomodal lognormal volume size ($r_d$) distribution, which is defined as follows:

$$\frac{dV(r_d)}{d \ln r_d} = \frac{C_v}{\sqrt{2\pi} \ln \sigma} \exp\left[-\frac{(\ln r_d - \ln r_v)^2}{2 \ln^2 \sigma}\right], \qquad \text{(A1)}$$

where $Cv$ is the particle volume concentration, $r_v$ is the volume median radius, and $\sigma$ is the standard deviation. $r_v$ is set to 0.143, 2.59, and 2.834 ($\sigma$ is 1.537, 2.054, and 1.908) for fine, coarse marine, and coarse dust, respectively, based on the observations by Omar et al. (2005) and Sayer et al. (2012). Regarding the aerosol shape, we assume a spherical model for the fine and coarse marine models, and a non-spherical model for the coarse dust model (Nakajima et al. 1989). The aerosol

vertical distribution is set to the same distribution that was used for rural (dominant at 0–2 km), sea-spray (below 2 km), and yellow sand (4-8 km), for fine, coarse marine, and coarse dust in the STAR code, respectively. The real part of the refractive index is set to 1.439, 1.362, and 1.452 for fine, coarse marine, and coarse dust, respectively, and the imaginary part of the refractive index ($m_i$) is set to $3.0\times10^{-9}$ and 0.0036 at all wavelengths for coarse marine, and coarse dust, respectively, based on Sayer et al. (2012) and Omar et al. (2005). The $m_i$ for the fine aerosol model is perturbed to represent non-absorbing and absorbing aerosols. To decrease the number of derived parameters, the $m_i$ for the fine aerosol model varies with change in $\eta_c^{dst}$ such that the fine and coarse models exhibit the same $\omega$ at 500 nm.

## Appendix B: Relationship of $\alpha$ and $\omega$ with $\eta_f$ and $\eta_c^{dst}$

Figure B1 shows the relations of the final retrieval parameters $\alpha$, and $\omega$ with the external mixing ratio of dry volume concentration of fine particles ($\eta_f$), and external mixing ratio of the dry volume concentration of dust particles for the coarse model ($\eta_c^{dst}$). The $\omega$ at 500 nm can be uniquely determined by the $\eta_c^{dst}$ (Fig. B1 (a)), since $\eta_c^{dst}$ for the coarse aerosol changes in conjunction with $m_i$ for the fine aerosol so that the $\omega$ at 500 nm has the same value without depending on the $\eta_f$. Note that the $\omega$ at wavelengths other than 500 nm are dependent on not only $\eta_c^{dst}$ but also $\eta_f$. The $\alpha$ is mainly determined by $\eta_f$, but also depends slightly on $\eta_c^{dst}$ (Fig. B1 (b)).

**Data availability**

Himawari-8/AHI aerosol data are available from JAXA Himawari Monitor site: https://www.eorc.jaxa.jp/ptree/index.html. The retrieved data in this study can be requested directly from the lead author (mayum@restec.or.jp).

**Author contributions**

MY developed retrieval code and analyzed the data with significant conceptual input from KY and critical technical support from TN, MK, and HM. KY and TT prepared the assimilated forecast data for test cases and long-time validations, respectively. MY prepared the manuscript with contributions from all co-authors.

**Competing interests**

The authors declare that they have no conflict of interest.

## Acknowledgements

The authors are grateful to the Open CLASTER project for allowing us to use the RSTAR package for this research. We would like to thank the AERONET project and its staff for establishing and maintaining the AERONET sites considered in this investigation. We also thank Dr. Haruma Ishida for providing the cloud detection algorithm (CLAUDIA). Finally, we appreciate the valuable discussions and support provided by Mr. Takashi Maki, Dr. Tsuyoshi Sekiyama, Dr. Makiko Hashimoto, and Prof. Teruyuki Nakajima. This study was supported by JSPS Grants-in-Aid for Scientific Research JP16H02946, and the JAXA/GCOM-C project.

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

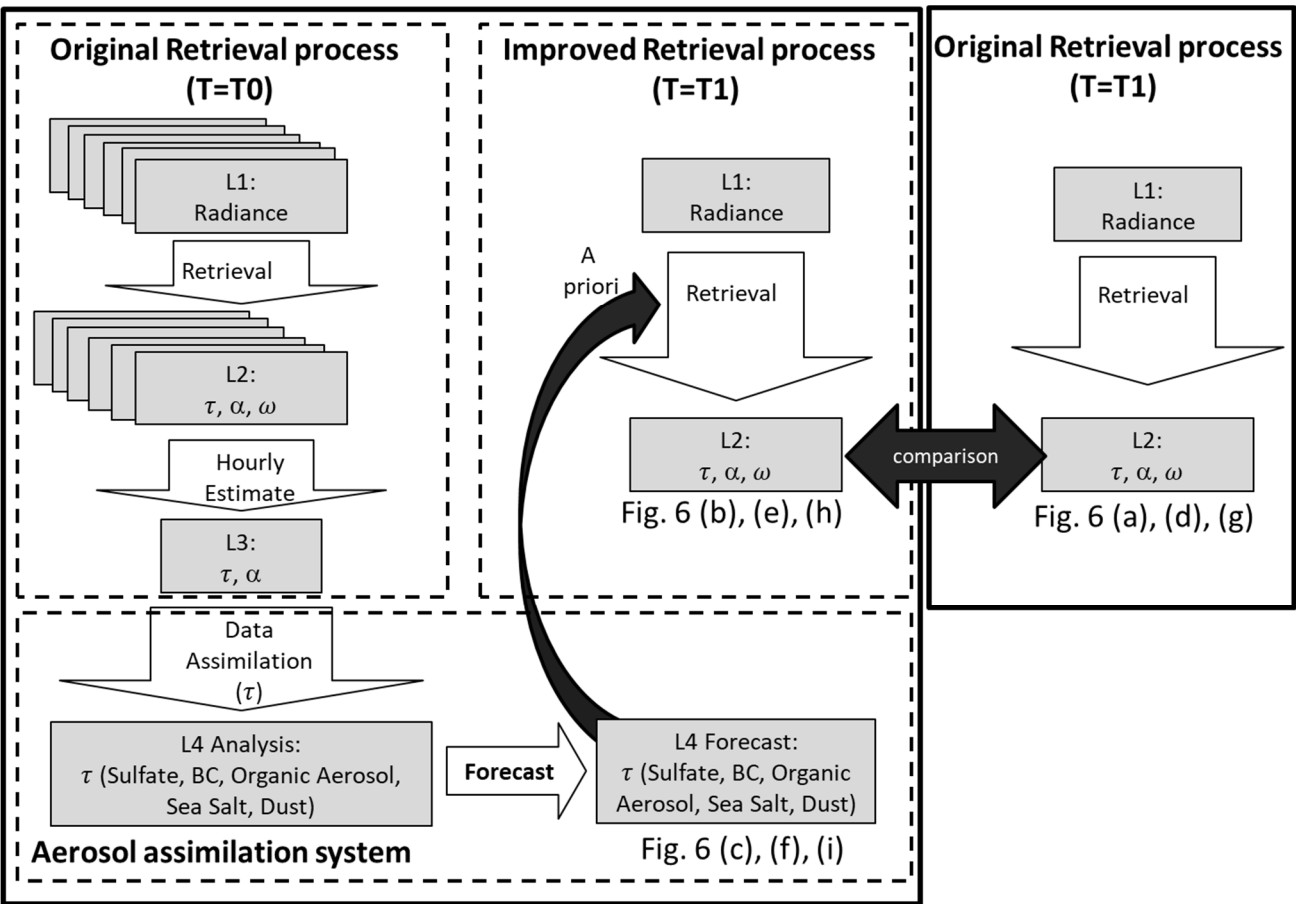

**Figure 1: Flowchart of data processing for aerosol retrieval at time T1.**

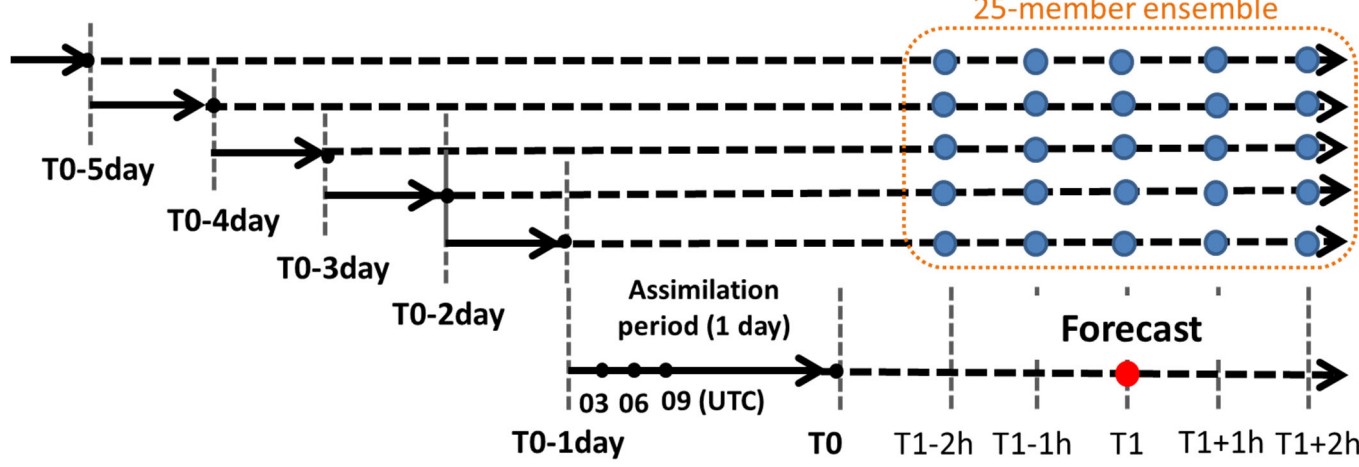


**Figure 2:** Forecast of aerosol transport model used for retrieval at time T1. Solid and dashed lines show the assimilation period (1 day) and forecast run, respectively.

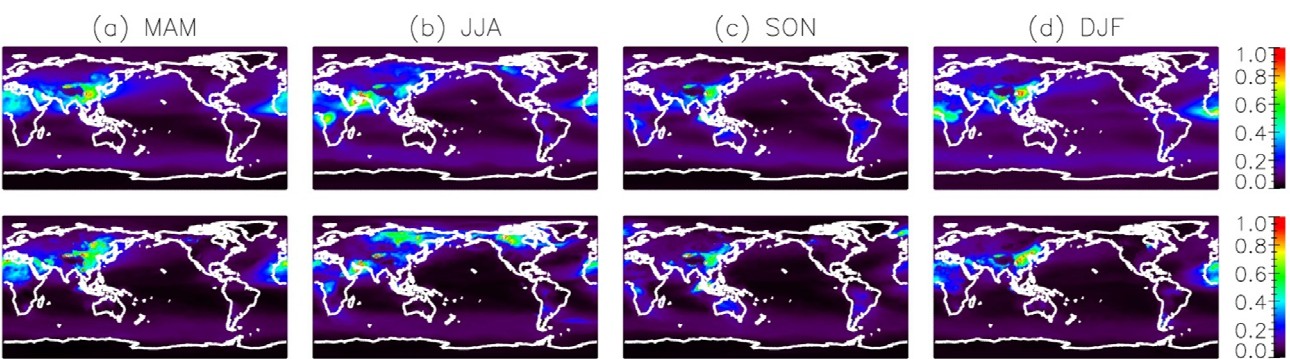

**Figure 3:** Mean (upper) and standard deviation (lower) of $\tau$ for free run model from 2011 to 2015 in (a) March, April and May, (b) June, July and August, (c) September, October and November, and (d) December, January and February.

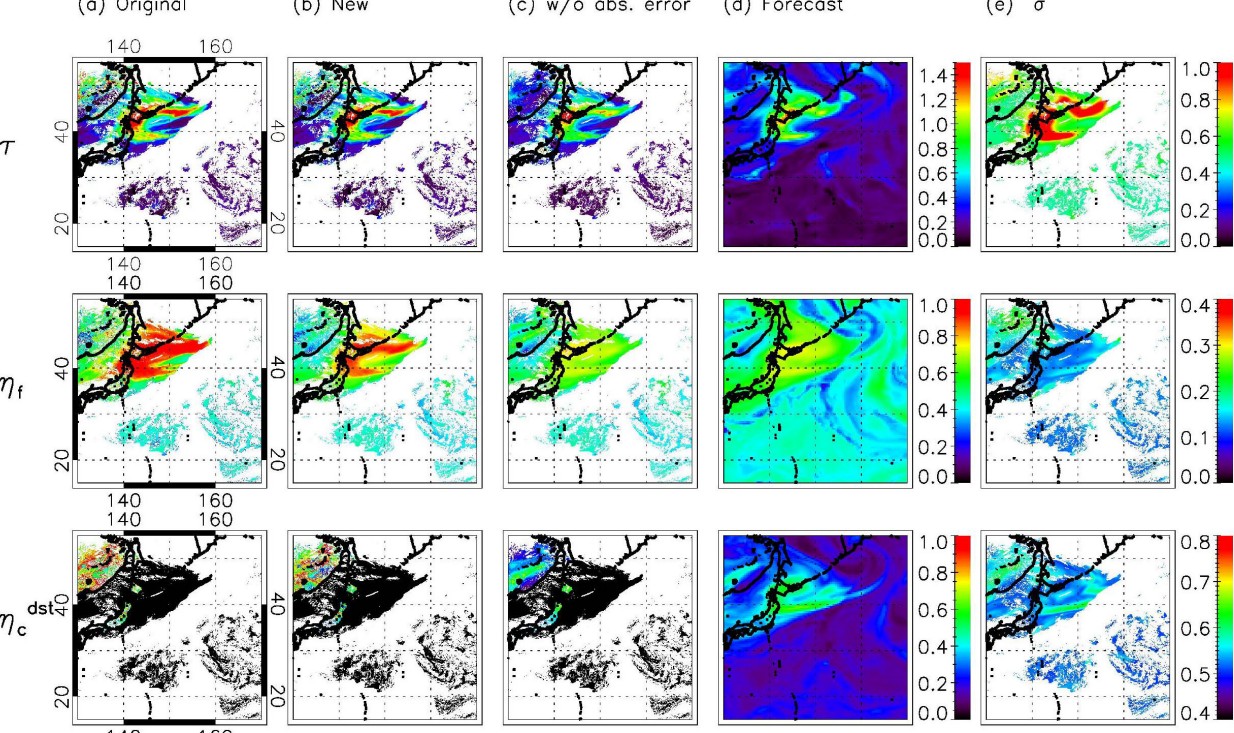

**Figure 4:** aerosol optical thickness at 500 nm $\tau$ (upper), external mixing ratio of dry volume concentration of fine particles $\eta_f$ (middle), and external mixing ratio of the dry volume concentration of dust particles for the coarse model $\eta_c^{dst}$ (lower) that are (a) retrieved from the original algorithm (i.e., using constant a priori estimate), (b) retrieved from this algorithm, (c) retrieved from this algorithm but without model absolute error ($S_a^A$), and (d) of the model forecast at 0200 UTC on May 19, 2016. (e) standard deviations of model forecast ($\sigma_{\tau_a}$, $\sigma_{\eta_{f_a}}$, and $\sigma_{\eta_c^{dst}}_a$) used for retrieval in (b).


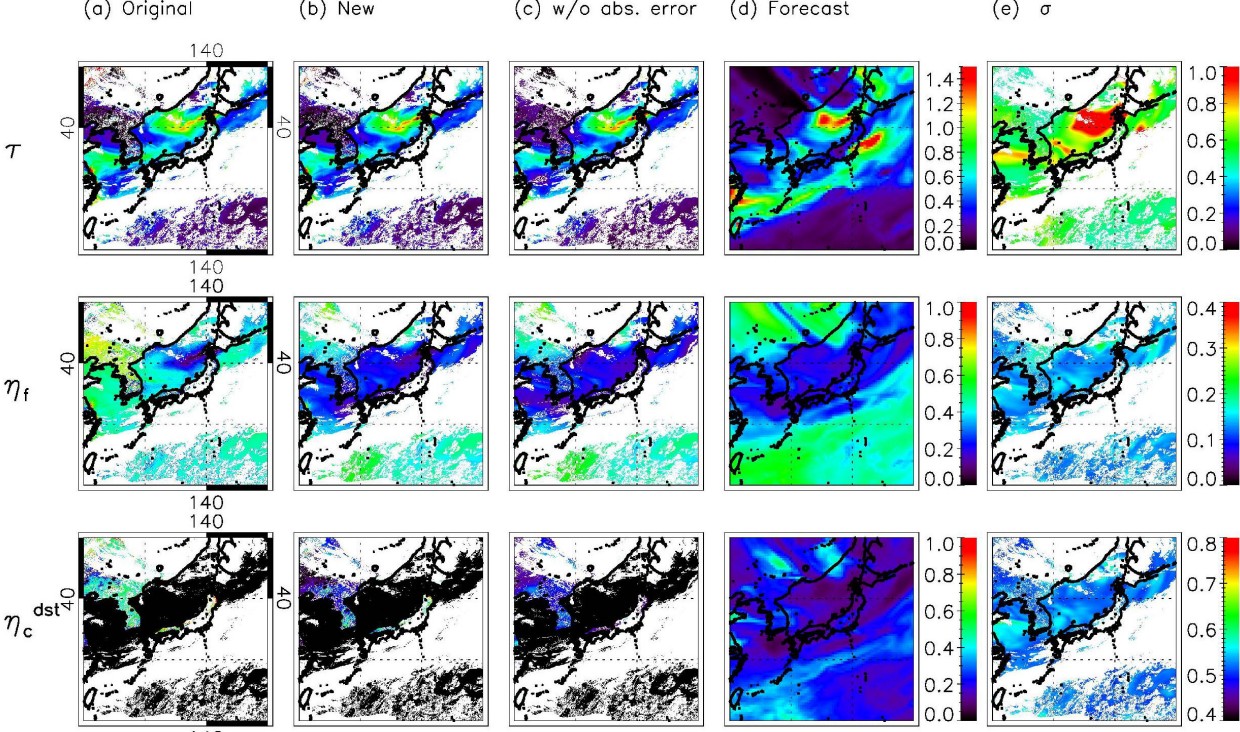

**Figure 5: Same as Fig. 4, except for the case at 0500 UTC on May 7, 2017.**

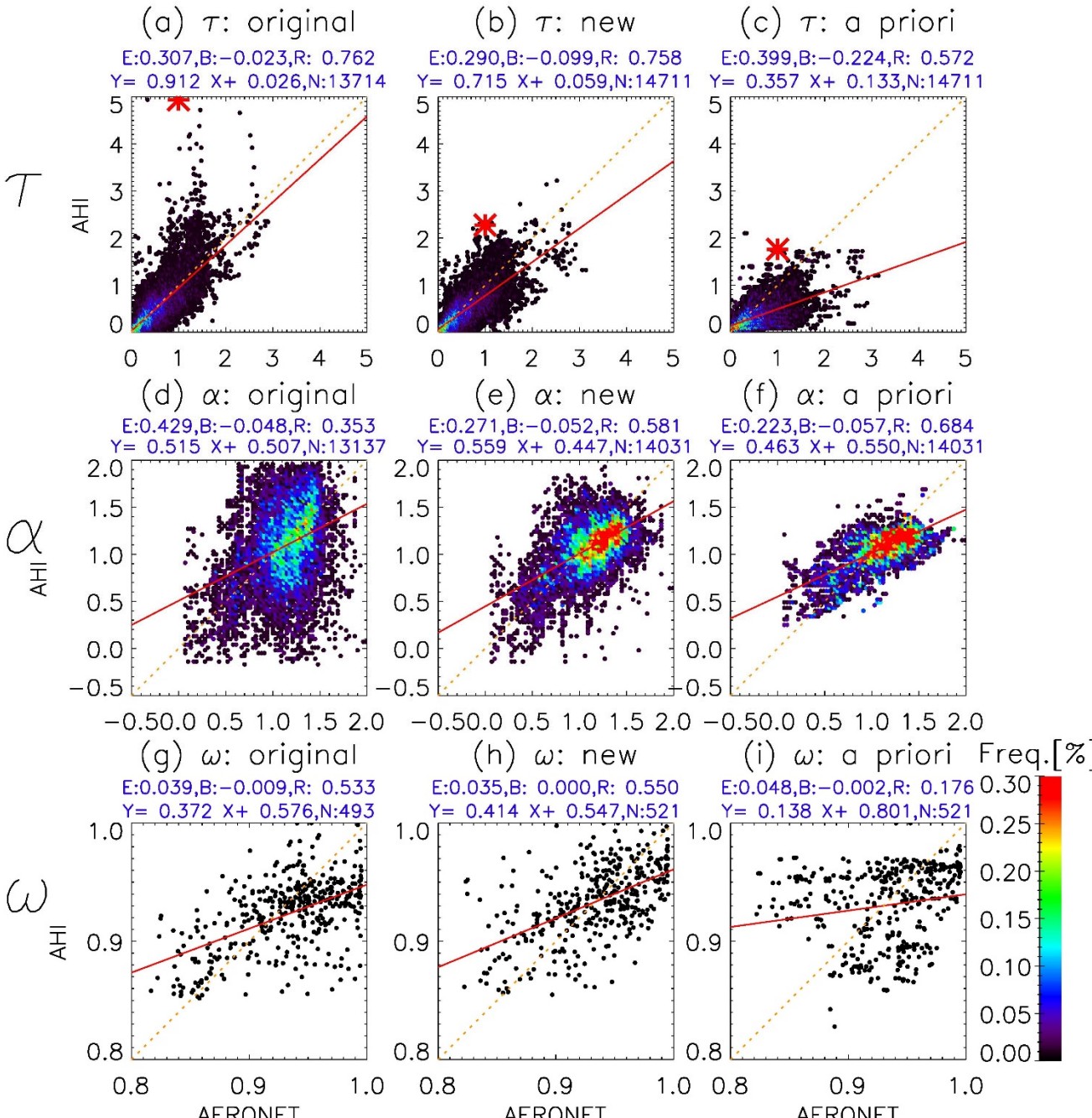

**Figure 6: Frequency distribution of $\tau$ (a, b, c), $\alpha$ (d, e, f), and $\omega$ (g, h, i) retrieved from AHI and those from AERONET. (a), (d), and (g) show the results from the original algorithm (i.e., using constant a priori), (b), (e), and (h) show the results from this algorithm, and (c), (f), and (i) are a priori estimate used for (b), (e), and (h), respectively. E, B, R, and N above the figures show the root mean square error, mean bias, correlation, and total number, respectively. Red asterisks in (a), (b), and (c) indicate the results at the red circles in Fig. 7 (a) and (b).**

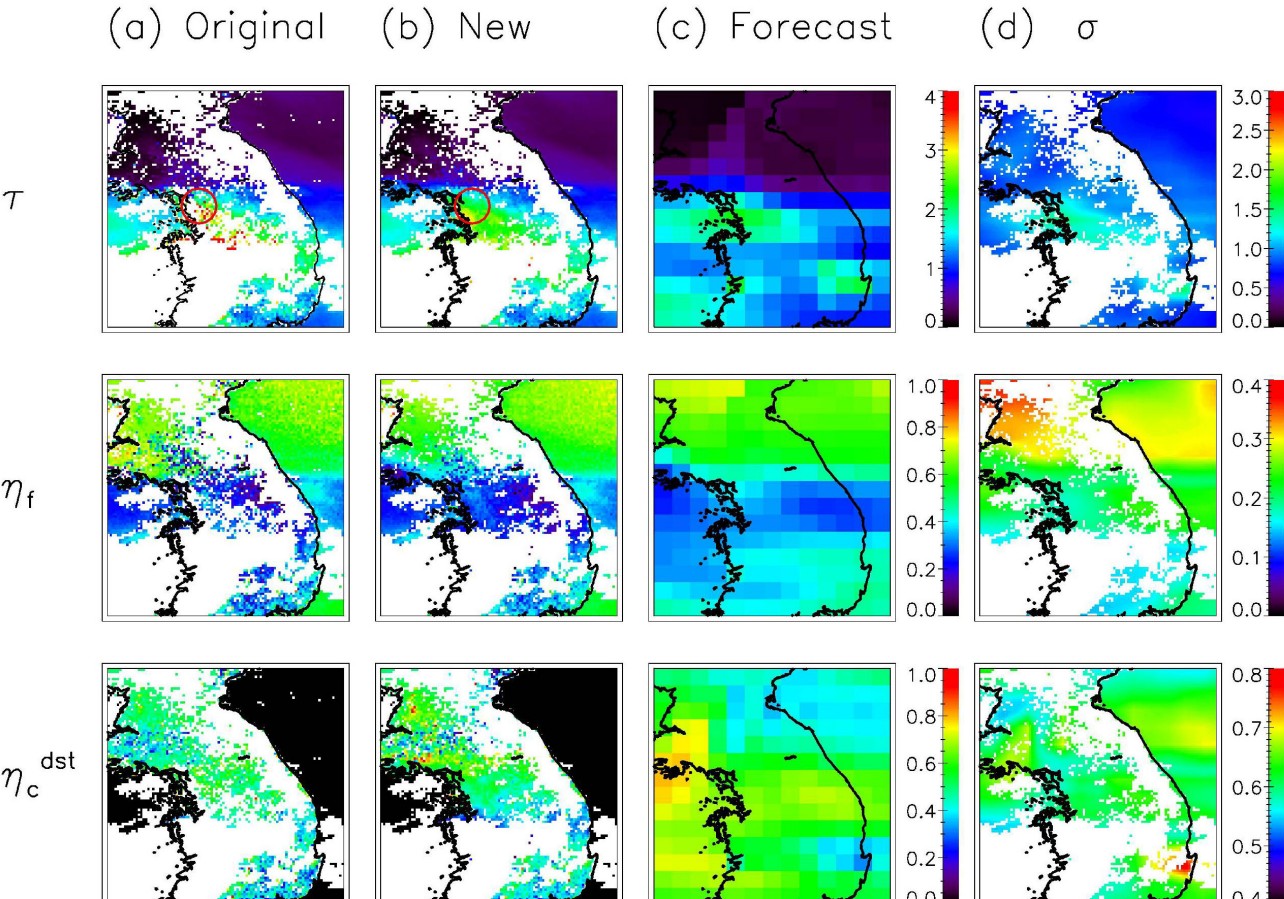

**Figure 7:** aerosol optical thickness at 500 nm $\tau$ (upper), external mixing ratio of dry volume concentration of fine particles $\eta_f$ (middle), and external mixing ratio of the dry volume concentration of dust particles for the coarse model $\eta_c^{dst}$ (lower) that are (a) retrieved from the original algorithm (i.e., using constant a priori estimate), (b) retrieved from this algorithm, and (c) from the model forecast at 0640 UTC on June 29, 2018. (d) standard deviations of model forecast ($\sigma_{\tau_a}$, $\sigma_{\eta_{f_a}}$, and $\sigma_{\eta_{c}^{dst}}{}_a$) used for retrieval in (b). Red circles in (a) $\tau$ and (b) $\tau$ indicate the results for the red asterisks in Fig. 6 (a), (b), and (c).


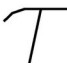

Figure 8 (panels a–i): Frequency distribution scatter plots of AHI vs AERONET.

(a) $\chi^2$:20.0, $S_\tau$:20.0
E:0.290,B:−0.099,R: 0.758
Y= 0.715 X + 0.059

(b) $\chi^2$:20.0, $S_\tau$: 1.0
E:0.276,B:−0.105,R: 0.774
Y= 0.716 X + 0.050

(c) $\chi^2$:20.0, $S_\tau$: 0.5
E:0.263,B:−0.110,R: 0.775
Y= 0.689 X + 0.048

(d) $\chi^2$: 0.5, $S_\tau$:20.0
E:0.274,B:−0.092,R: 0.775
Y= 0.746 X + 0.050

(e) $\chi^2$: 0.5, $S_\tau$: 1.0
E:0.263,B:−0.096,R: 0.787
Y= 0.743 X + 0.046

(f) $\chi^2$: 0.5, $S_\tau$: 0.5
E:0.252,B:−0.100,R: 0.792
Y= 0.726 X + 0.043

(g) $\chi^2$: 0.2, $S_\tau$:20.0
E:0.259,B:−0.085,R: 0.789
Y= 0.765 X + 0.041

(h) $\chi^2$: 0.2, $S_\tau$: 1.0
E:0.247,B:−0.086,R: 0.799
Y= 0.764 X + 0.038

(i) $\chi^2$: 0.2, $S_\tau$: 0.5
E:0.233,B:−0.083,R: 0.800
Y= 0.750 X + 0.039

Freq.[%]
0.30
0.25
0.20
0.15
0.10
0.05
0.00

AHI (y-axis)
AERONET (x-axis)

**Figure 8: Frequency distribution of $\tau$ retrieved from AHI and those from AERONET. The results retrieved from this algorithm in the case of $\chi^2$ less than 20, 0.5, 0.2, and uncertainties of the retrieved $\tau$ ($S_\tau$) less than 20, 1.0, 0.5 are plotted in each panel. E, B, and R above the figures show the root mean square error, mean bias, and correlation, respectively.**


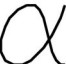

Figure 9: Frequency distribution of $\alpha$ retrieved from AHI and those from AERONET. The results retrieved from this algorithm in the case of $\chi^2$ less than 20, 0.5, 0.2, and uncertainties of the retrieved $\eta_f$ ($S_{\eta_f}$) less than 20, 0.5, 0.2 are plotted in each panel. E, B, and R above the figures are the same as in Fig. 8.

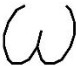

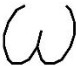

(a) $\chi^2$:20.0, $S_{\eta_{c^{dst}}}$:20.0
E:0.035,B: 0.000,R: 0.550
Y= 0.414 X + 0.547

(b) $\chi^2$:20.0, $S_{\eta_{c^{dst}}}$: 0.5
E:0.037,B: 0.001,R: 0.516
Y= 0.395 X + 0.557

(c) $\chi^2$:20.0, $S_{\eta_{c^{dst}}}$: 0.2
E:0.033,B:−0.005,R: 0.548
Y= 0.495 X + 0.456

(d) $\chi^2$: 0.5, $S_{\eta_{c^{dst}}}$:20.0
E:0.035,B: 0.004,R: 0.575
Y= 0.420 X + 0.545

(e) $\chi^2$: 0.5, $S_{\eta_{c^{dst}}}$: 0.5
E:0.035,B: 0.009,R: 0.596
Y= 0.478 X + 0.488

(f) $\chi^2$: 0.5, $S_{\eta_{c^{dst}}}$: 0.2
E:0.031,B: 0.002,R: 0.599
Y= 0.561 X + 0.402

(g) $\chi^2$: 0.2, $S_{\eta_{c^{dst}}}$:20.0
E:0.035,B: 0.005,R: 0.573
Y= 0.413 X + 0.551

(h) $\chi^2$: 0.2, $S_{\eta_{c^{dst}}}$: 0.5
E:0.035,B: 0.010,R: 0.577
Y= 0.460 X + 0.504

(i) $\chi^2$: 0.2, $S_{\eta_{c^{dst}}}$: 0.2
E:0.031,B: 0.003,R: 0.548
Y= 0.478 X + 0.478


**Figure 10: Frequency distribution of $\omega$ retrieved from AHI and those from AERONET. The results retrieved from this algorithm in the case of $\chi^2$ less than 20, 0.5, 0.2, and uncertainties of the retrieved $\eta_c^{dst}$ ($S_{\eta_c^{dst}}$) less than 20, 0.5, 0.2 are plotted in each panel. E, B, and R above the figures are the same as in Fig. 8.**

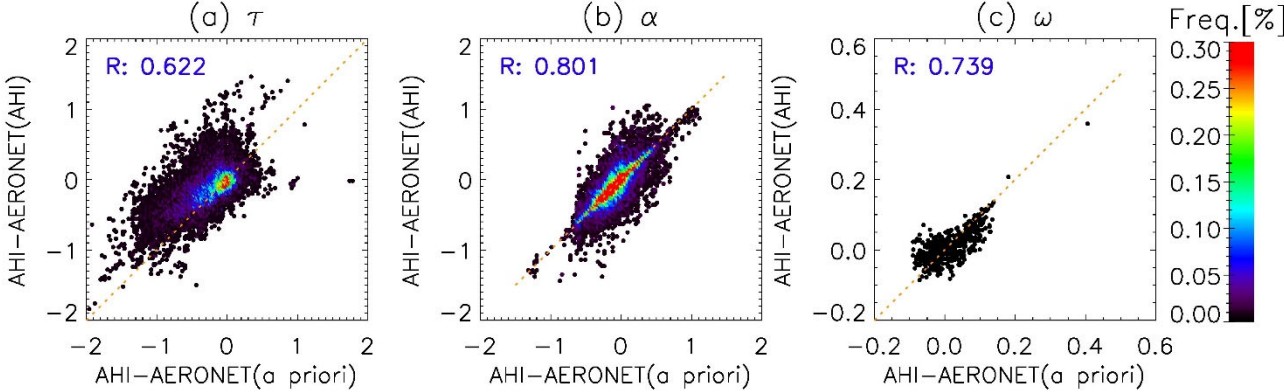


**Figure 11: Frequency distribution of the difference between $\tau$ (a), $\alpha$ (b), and $\omega$ (c) retrieved from AHI and those from AERONET, as a function of the difference between $\tau$ (a), $\alpha$ (b), and $\omega$ (c) of a priori estimate and AERONET. R shows the correlation.**


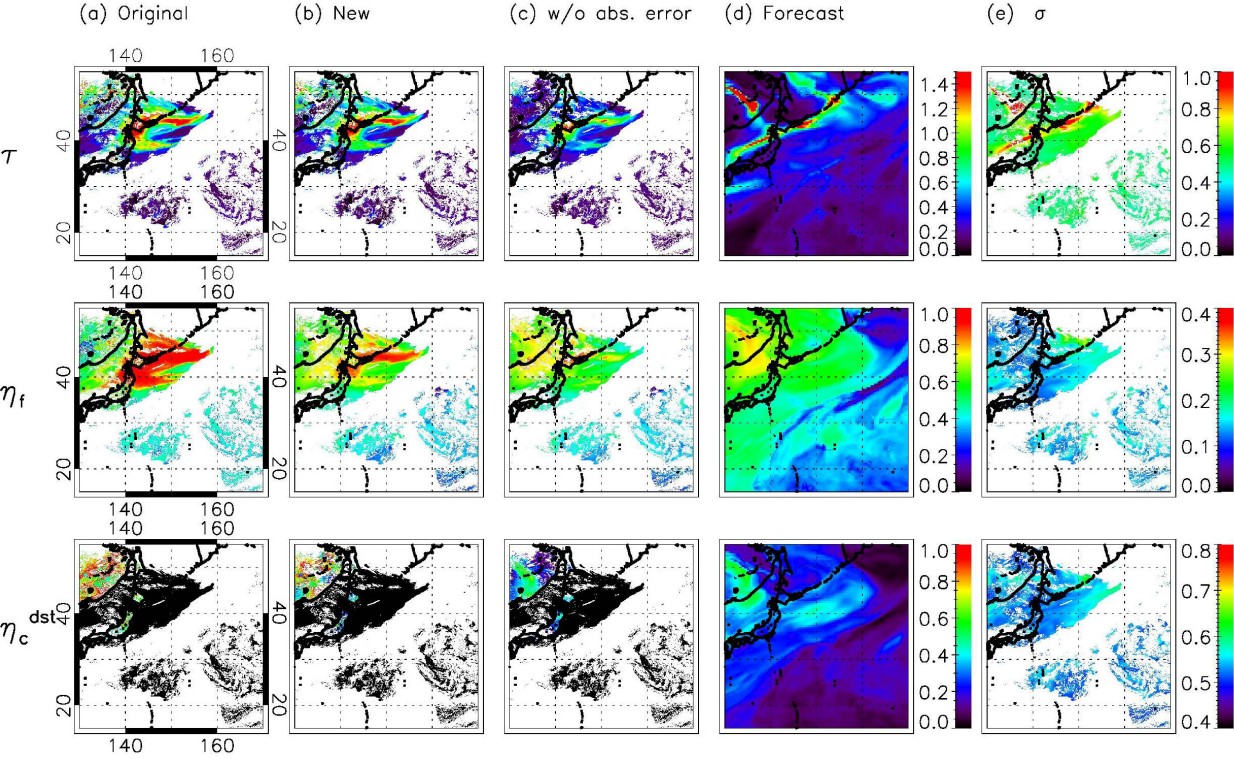

**Figure 12: Same as Fig. 4, except for using the forecast on April 27, 2018 as a priori estimate.**

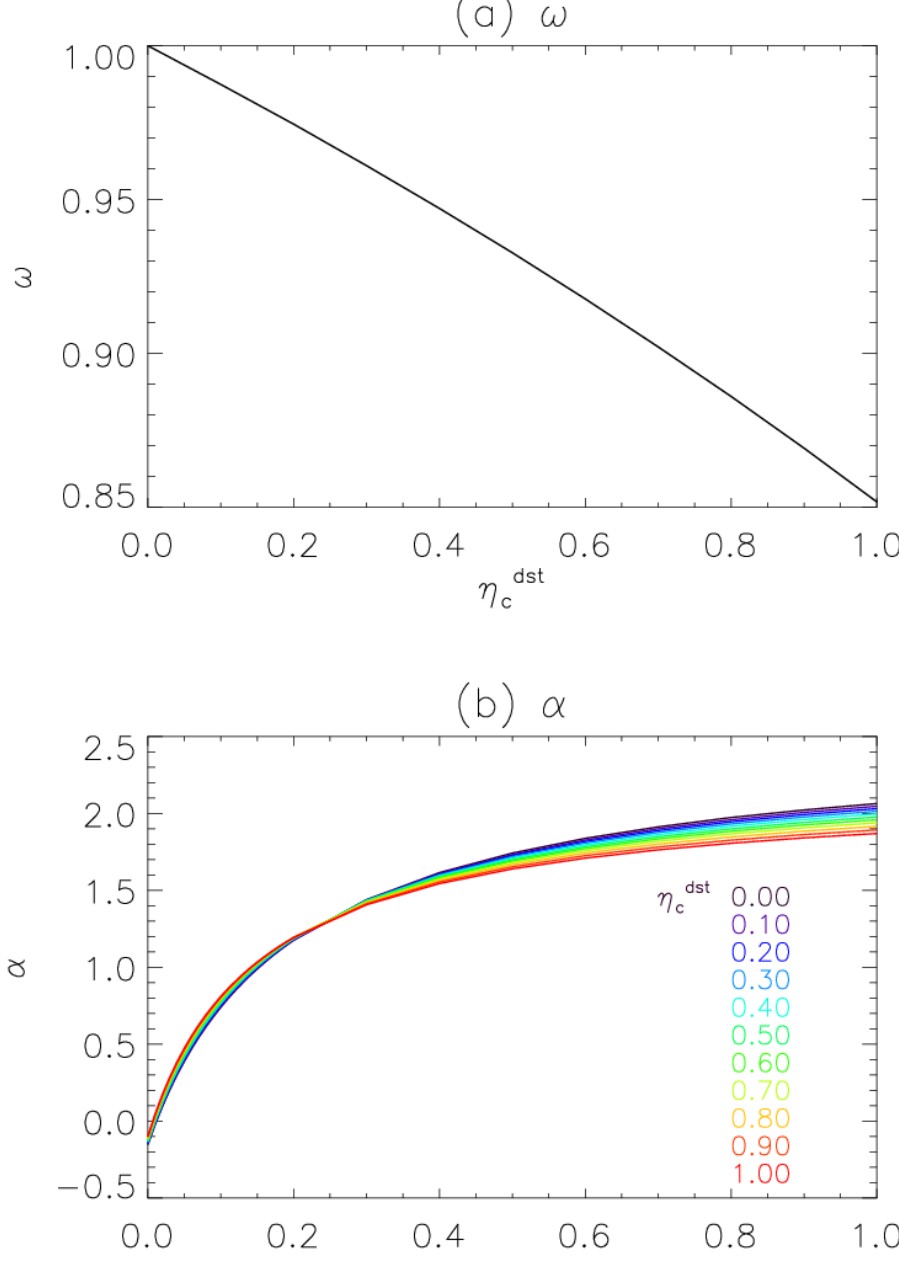


**Figure B1: The relations of (a) single-scattering albedo at 500 nm ($\omega$), and (b) Ångström exponent between 400 and 600 nm ($\alpha$) with the external mixing ratio of dry volume concentration of fine particles ($\eta_f$), and external mixing ratio of the dry volume concentration of dust particles for the coarse model ($\eta_c^{dst}$). Each color represents a different $\eta_c^{dst}$.**