# Peer review of "Retrieval of Aerosol Combined with Assimilated Forecast"

_Atmospheric Chemistry and Physics, 2020_

## Referee Comment (RC1) · Alexander Kokhanovsky (Referee) · 2 Jun 2020

This work is aimed to the retrieval of aerosol properties such as aerosol optical thickness (AOT), Angstrom exponent(AE) and single scattering albedo (SSA) using satellite observations. The authors use the short/term forecats from an aerosol assimilation system for a priori estimate of parameters to be retrieved. I would suggest the publication of this paper taking into account the comments given below: 1). Please, give equations related the parameters AOT, AE, SSA with fine particles mixing ratio and the imaginary part of their refractive index. Please, specify all assumptions used to derive the corresponding relationships. 2). Please, extend the discussion of Fig.6 pointing to the reasons for possible large deviations of ground and satellite derived aerosol products for the case of developed algorithm.

---

## Referee Comment (RC2) · Anonymous Referee #3 · 1 Sep 2020

The paper "Retrieval of Aerosol Combined with Assimilated Forecast" is an interesting paper that uses model forecast to improve the aerosol satellite retrieval especially over aerosol absorption and size parameters. However, there are couple major problems need to be clarified. First of all, what is the relations between Angstrom exponent and particle mixing ratio? If this ratio is similar to fine mode fraction, which defined as fine mode AOD over total AOD, then the Angstrom Exponent is not only depending on this parameter. Similarly, the single scattering albedo is not only dependent on imaginary part of the refractive index, it is also function of size distribution. Also in the paper, it claims that imaginary part of the refractive index is between 0-1, but realistically, this value is between 0.00001 to 0.01 at 550nm. Look at the value the author cited in line 36 these values are around e-8. So I am not sure how is this value in case study can

be above 0.1 (which wavelength are we talking about here)? Is mi really the imaginary part of the refractive index? Also setting aerosol to non-absorbing totally causes big problem when there is transported dust/smoke over ocean. The case study shown the improved relation of Angstrom exponent/single scattering albedo vs ground truth, I think it is worth exploring with more cases with more discussion of the error sources in each cases.

---

## Author Comment (AC1) · 15 Nov 2020

**Response to reviewers**

**Title: Satellite Retrieval of Aerosol Combined with Assimilated Forecast**
**Authors: M. Yoshida et al.**
**MS No.: acp-2020-356**

**Response to Dr. Alexander Kokhanovsky (Referee #1)**

We would like to thank Dr. Alexander Kokhanovsky for the constructive comments and recommendations for publication. We modified the manuscript accordingly, and we believe the revised paper is improved. Our point-by-point responses and actions regarding the comments are listed below. The comments from the reviewers are *emphasized*, and our responses and actions are shown in blue. Modified parts in the revised manuscript are shown in red. English correction by several native speakers is shown in green. The original sentences removed in the revised manuscript are shown in orange.

*This work is aimed to the retrieval of aerosol properties such as aerosol optical thickness(AOT),Angstromexponent(AE)andsinglescatteringalbedo(SSA)usingsatellite observations. The authors use the short/term forecats from an aerosol assimilation system for a priori estimate of parameters to be retrieved. I would suggest the publication of this paper taking into account the comments given below:*
*1). Please, give equations related the parameters AOT, AE, SSA with fine particles mixing ratio and the imaginary part of their refractive index. Please, specify all assumptions used to derive the corresponding relationships.*

Thank you very much for your comment. We are sorry for the inadequate explanation. We added the detailed explanation of the aerosol model (appendix A), and the relationship of $\alpha$ and $\omega$ with $\eta_f$ and $\eta_c^{dst}$ (appencndix B) in the revised manuscript. We also added the Figure B1, which shows the relations of (a) $\omega$ and (b) $\alpha$ with $\eta_f$ and $\eta_c^{dst}$ as follows.

[P4_L112]

[revised manuscript text omitted]

*2). Please, extend the discussion of Fig.6 pointing to the reasons for possible large deviations of ground and satellite derived aerosol products for the case of developed algorithm.*

Thank you very much for your valuable comment. We investigate the cause of the possible large deviation of validation results of this algorithm, and added Fig.8, 9, 10, 11 in the revised manuscript. Here, the validation results in Figs.6, 8, 9, 10, 11 are revised for the extended six months in response to Reviewers#3 comments.

[P8_L247]

We also investigated the cause of the possible large deviation between the retrieved parameters from the new algorithm and the ground observation. Figures 8, 9, and 10 show the validation results of $\tau$, $\alpha$, and $\omega$, respectively, when the chi-square value ($\chi^2$) and the uncertainties of the retrieved three parameters ($\tau$, $\eta_f$, and $\eta_c^{dst}$) are smaller than a threshold. The chi-square value ($\chi^2$) is calculated as follows:

$$\chi^2[\boldsymbol{R} - \boldsymbol{F}(\boldsymbol{x})] = [\boldsymbol{R} - \boldsymbol{F}(\boldsymbol{x})]^T \boldsymbol{S}_e^{-1}[\boldsymbol{R} - \boldsymbol{F}(\boldsymbol{x})]/N, \quad (7)$$

and shows the closeness of the retrieved value to the observed value. The covariance matrix of the uncertainties of the retrieved parameters $\boldsymbol{S}_{\hat{x}}$ is calculated using the law of error propagation, as follows:

$$\boldsymbol{S}_{\hat{x}} = (\boldsymbol{A}^T \boldsymbol{S}_e^{-1} \boldsymbol{A})^{-1} , (8)$$

where $\boldsymbol{A}$ is the Jacobian matrix. $\boldsymbol{S}_e$ is the covariance matrix of $\boldsymbol{R}$, and calculated from sum of sensor noise and the uncertainty in TOA reflectance that results from surface reflectance uncertainty (Yoshida et al., 2018). In reality, the $\boldsymbol{S}_e$ is almost determined by the uncertainty in TOA reflectance that results from surface reflectance uncertainty, because sensor noise is much smaller. Therefore, the $\boldsymbol{S}_{\hat{x}}$ is mostly caused by the surface reflectance uncertainty. Figure 8 shows that RMSE for $\tau$ decreases as the threshold of $\chi^2$ or $\boldsymbol{S}_{\hat{x}}$ becomes strict (i.e., decreases). On the other hand, RMSE for $\alpha$ (in Fig. 9) is not dependent on the threshold of $\boldsymbol{S}_{\hat{x}}$, but decreases as the $\chi^2$ threshold decreases. RMSE for $\omega$ (in Fig. 10) is little dependent on the threshold of $\boldsymbol{S}_{\hat{x}}$ and $\chi^2$. Next, in Fig. 11 we investigated how the retrieved accuracy (difference between aerosol parameters retrieved from AHI and those of AERONET) depends on the model's (i.e., a priori) accuracy. The retrieved accuracy of $\alpha$ and $\omega$ has strong linear relationships (a correlation of 0.801, and 0.739, respectively) to the model's accuracy, while that of $\tau$ has a moderate linear relationship (a correlation of 0.622). Summarizing these results, the retrieved accuracy of $\tau$ depends on all of the closeness to the observed value, accuracy of the surface reflectance estimation, and accuracy of a priori estimate, while the accuracy of a priori estimate is critical for the retrieved accuracy of $\alpha$ and $\omega$. Thus, introducing more realistic a priori estimates in the new retrieval algorithm instead of the constant values in the original algorithm led to the improvement of RMSE. It is also shown that the improvement of a numerical aerosol forecast by improving the aerosol transport model and the assimilation method, and increasing the assimilation frequency may further improve the retrieval

accuracy in the future.

[Figure]

**Figure 8: Frequency distribution of $\tau$ retrieved from AHI and those from AERONET. The results retrieved from this algorithm in the case of $\chi^2$ less than 20, 0.5, 0.2, and uncertainties of the retrieved $\tau$ ($S_\tau$) less than 20, 1.0, 0.5 are plotted in each panel. E, B, and R above the figures show the root mean square error, mean bias, and correlation, respectively.**

[Figure]

**Figure 9:** Frequency distribution of α retrieved from AHI and those from AERONET. The results retrieved from this algorithm in the case of $\chi^2$ less than 20, 0.5, 0.2, and uncertainties of the retrieved $\eta_f$ ($S_{\eta_f}$) less than 20, 0.5, 0.2 are plotted in each panel. E, B, and R above the figures are the same as in Fig. 8.

[Figure]

**Figure 10: Frequency distribution of $\omega$ retrieved from AHI and those from AERONET. The results retrieved from this algorithm in the case of $\chi^2$ less than 20, 0.5, 0.2, and uncertainties of the retrieved$\eta_c^{dst}$ ($S_{\eta_c^{dst}}$) less than 20, 0.5, 0.2 are plotted in each panel. E, B, and R above the figures are the same as in Fig. 8.**

[Figure]

**Figure 11: Frequency distribution of the difference between $\tau$ (a), $\alpha$ (b), and $\omega$ (c) retrieved from AHI and those from AERONET, as a function of the difference between $\tau$ (a), $\alpha$ (b), and $\omega$ (c) of a priori estimate and AERONET. R shows the correlation.**

**Response to Refree #3**

We would like to thank Anonymous Refree #3 for the constructive comments and recommendations for publication. We modified the manuscript accordingly, and we believe the revised paper is improved. Our point-by-point responses and actions regarding the comments are listed below. The comments from the reviewers are *emphasized*, and our responses and actions are shown in blue. Modified parts in the revised manuscript are shown in red. English correction by several native speakers is shown in green. The original sentences removed in the revised manuscript are shown in orange.

*The paper "Retrieval of Aerosol Combined with Assimilated Forecast" is an interesting paper that uses model forecast to improve the aerosol satellite retrieval especially over aerosol absorption and size parameters. However, there are couple major problems need to be clarified. First of all, what is the relations between Angstrom exponent and particle mixing ratio? If this ratio is similar to fine mode fraction, which defined as fine mode AOD over total AOD, then the Angstrom Exponent is not only depending on this parameter. Similarly, the single scattering albedo is not only dependent on imaginary part of the refractive index, it is also function of size distribution.*

Thank you very much for your comment. The $\eta_f$ is the external mixing ratio of the dry volume concentration of fine particles. For the coarse aerosol model, we set the external mixture ($\eta_c^{dst}$ is the mixing ratio) of a pure marine aerosol and a dust model. For the fine aerosol model, the imaginary part of the refractive index $m_i$ was perturbed to represent a non-absorbing and absorbing aerosol. To decrease the number of derived parameters, the $m_i$ varied with $\eta_c^{dst}$ such that the fine and coarse models exhibited the same $\omega$ at 500 nm.

As you pointed out the angstrom Exponent $\alpha$ is not only depending on the $\eta_f$, but also depending on $\eta_c^{dst}$. We modified the following sentence in the revised manuscript.

[P5_L155]
$\sigma_{\eta_{f_a}}^A$ (0.093) is calculated from RMSE of the model's $\alpha$ (0.223 in Fig. 6 (f)) at $\alpha$ of 1.2 and $\eta_c^{dst}$ of 0.5.
The original manuscript was:
$\sigma_{\eta_{f_a}}^A$ (0.110) is calculated from RMSE of the model's $\alpha$ (0.233 in Fig. 6 (f)), which is uniquely determined by $\eta_f$.

The single scattering albedo $\omega$ is function of $\eta_c^{dst}$(or $m_i$) and $\eta_f$, but $\omega$ at 500nm can be uniquely determined by $\eta_c^{dst}$(or $m_i$), since $\eta_c^{dst}$ moves in conjunction with $m_i$ so that the $\omega$ at 500 nm has the same value without depending on the $\eta_f$. We added the detailed explanation of the aerosol model (appendix A), and the relationship of α and $\omega$ with $\eta_f$ and $\eta_c^{dst}$ (appendix B) in the revised manuscript. We also added the Figure B1, which shows the relations of (a) $\omega$ and (b) α with $\eta_f$ and $\eta_c^{dst}$ as follows.

**[P4_L107]**

[revised manuscript text omitted]

*Also in the paper, it claims that imaginary part of the refractive index is between 0-1, but realistically, this value is between 0.00001 to 0.01 at 550nm. Look at the value the author cited in line36 these values are around e-8. So I am not sure how is this value in case study can be above 0.1 (which wavelength are we talking about here)? Is mi really the imaginary part of the refractive index?*

We truly appreciate your effort to read our paper carefully. We are sorry that the $m_i$ was a mistake of $\eta_c^{dst}$. We modified from $m_i$ to $\eta_c^{dst}$ in the revised manuscript. Here, the qualitative characteristics shown in the manuscript do no change, since the $m_i$ varied with $\eta_c^{dst}$ such that the fine and coarse models exhibited the same $\omega$ at 500 nm.

[P4_L107]

In the retrieval process, the final retrieval parameters ( $\tau$, $\alpha$ , and $\omega$ ) are calculated from the set of aerosol parameters ($\tau$, external mixing ratio of dry volume concentration of fine particles $\eta_f$, and external mixing ratio of the dry volume concentration of dust particles for the coarse model $\eta_c^{dst}$) defined by Yoshida et al. (2018). Here, the imaginary part of the refractive index ($m_i$) for the fine aerosol model varies with change in $\eta_c^{dst}$ such that the fine and coarse models exhibit the same $\omega$ at 500 nm (see Yoshida et al., 2018 for more details).

*Also setting aerosol to non-absorbing totally causes big problem when there is transported dust/smoke over ocean.*

Thank you very much for your valuable comment. We tried to use model's $\eta_c^{dst}$ as a priori over ocean, in order to handle the absorbing aerosol over ocean, but using the model's $\eta_c^{dst}{}_a$ as $\eta_c^{dst}$ over the ocean sometimes leads to a large $\tau$ estimation, since the model's $\eta_c^{dst}$ over ocean is not good at this time. We we will use the model's $\eta_c^{dst}{}_a$ as $\eta_c^{dst}$ over the ocean after obtaining a better model $\eta_c^{dst}{}_a$ in the future. In the revised manuscript, we clearly added this problem as follows.

[P6_L166]

Therefore, $\eta_c^{dst}$ over the ocean, which is the least sensitive to satellite observation, is set to 0 (i.e., non-absorbing aerosol) at this time, because the aerosol over the ocean is generally less absorbing than that over land, and using the model's $\eta_c^{dst}{}_a$ as $\eta_c^{dst}$ over the ocean leads to a worse estimation of $\tau$ (not shown). However, using non-absorbing aerosol over ocean causes a big problem in case of dust/smoke transported over the ocean, so we will use the model's $\eta_c^{dst}{}_a$ as $\eta_c^{dst}$ over the ocean after obtaining a better model's $\eta_c^{dst}{}_a$ in the future. Note that $\eta_c^{dst}$ over land is properly retrieved from satellite data (i.e., not set to 0) using the model's $\eta_c^{dst}{}_a$ as a priori estimate, since the number of satellite channels (five) used over land is greater than the number of retrieval parameters (three).

*The case study shown    the improved relation of Angstrom exponent/single scattering albedo vs ground truth, I    think it is worth exploring with more cases with more discussion of the error sources in    each cases.*

Thank you very much for your constructive comment. We increased the case by extending the validation period from three months to six months.

**[P7_L206]**

Initial validation was conducted for six months (March, April, May, June, July, 2018, and February 2019). Long-term validation will be required in future studies.

In addition, we investigated the appropriate setting for $S_a$ by validating the results for various conditions, and change slightly the setting for $S_a$.

The original manuscript used the following $S_a$.

**[P6_L2]**

we assume that non-diagonal component of $\boldsymbol{S_a}$ and $\boldsymbol{S_a^E}$ is the same, and calculate the non-diagonal component of $\boldsymbol{S_a}$ as follows:

$$\sigma_{\tau_a \eta_{f_a}} = \frac{\sigma_{\tau_a} \cdot \sigma_{\eta_{f_a}}}{\sigma_{\tau_a}^E \cdot \sigma_{\eta_{f_a}}^E} \sigma_{\tau_a \eta_{f_a}}^E, \qquad (7)$$

$$\sigma_{\tau_a \eta_c^{dst}{}_a} = \frac{\sigma_{\tau_a} \cdot \sigma_{\eta_c^{dst}{}_a}}{\sigma_{\tau_a}^E \cdot \sigma_{\eta_c^{dst}{}_a}^E} \sigma_{\tau_a \eta_c^{dst}{}_a}^E, \qquad (8)$$

$$\sigma_{\eta_{f_a} \eta_c^{dst}{}_a} = \frac{\sigma_{\eta_{f_a}} \cdot \sigma_{\eta_c^{dst}{}_a}}{\sigma_{\eta_{f_a}}^E \cdot \sigma_{\eta_c^{dst}{}_a}^E} \sigma_{\eta_{f_a} \eta_c^{dst}{}_a}^E. \qquad (9)$$

The revised manuscript uses the following $S_a$.

we use the non-diagonal components of $\boldsymbol{S_a^E}$ as those of $\boldsymbol{S_a}$.

With the above changes, and minor bug fixes, Fig.4, 5, 6, 7, 8 (12 in the revised manuscript) changed slightly. The major results are not changed, but the numerical values were changed as follows.

**[P7_L218]**

For the $\tau$ estimations (Fig. 6 (a), (b), (c)), the root mean square error (RMSE), mean bias (MB), and correlation (0.290, -0.099, and 0.758) from this algorithm (Fig. 6 (b)) are all better than those (0.399, -0.224, and 0.572) from the model forecast (i.e., a priori estimate) in Fig. 6 (c), which means that satellite information is very effective for the retrieval of $\tau$. In addition, the RMSE (0.290) in Fig. 6 (b) is better than that (0.307) without the forecast model (Fig. 6 (a)), which means that the model information is also effective and the improved algorithm shows better performance than the original algorithm. The MB

(-0.099) in Fig. 6 (b) is worse than that (-0.023) in Fig. 6 (a), probably because the large outlier in Fig. 6 (a) is improved in Fig. 6 (b).

**[P7_L230]**

For the $\alpha$ estimations (Fig. 6 (d), (e), (f)), large variance in the original method is considerably reduced by this method. The RMSE and correlation (0.271 and 0.581) from this algorithm (Fig. 6 (e)) are much better than those (0.429 and 0.353) from the original algorithm without the forecast model (Fig. 6 (d)), which indicates that the new algorithm could improve the precision of $\alpha$ estimations by adding more accurate $\alpha$ (RMSE of 0.223) information from the model. In addition, the MB (-0.052) from this algorithm (Fig. 6 (e)) is better than that (-0.057) from the model forecast (Fig. 6 (f)), due to the improvement of negative bias in the large $\alpha$ in the model forecast.

**[P7_L238]**

For the $\omega$ estimations (Fig. 6 (g), (h), (i)), the RMSE, MB, and correlation (0.032, -0.004, and 0.530) from this algorithm (Fig. 6 (h)) are better than those (0.046, -0.014, and 0.218) from the model forecast (Fig. 6 (i)), which indicates the effectiveness of satellite information for $\omega$ retrieval.
The following sentence

"In addition, while statistic scores (i.e., RMSD, correlation, MB) show little modification, this algorithm improved the slope and intercept of the regression line by introducing the model forecast. This is probably because the model's $\omega$ is not very consistent with AERONET (RMSE of 0.053), but less biased (-0.001 of MB) due to the possibility that the model's $\omega$, whose determinants are complex (e.g., different $\omega$ for the same type of aerosol), generally reflects reality, but not enough in individual cases."

was changed to

In addition, this algorithm improved RMSD, MB, and correlation by introducing the model forecast.

We also added the comparison of the total number of validation points and successfully retrieved area.

**[P7_L210]**

The validation of $\alpha$ and $\omega$ is limited to cases where the simultaneously retrieved $\tau$ are greater than 0.3 because there is little information of $\alpha$ and $\omega$ from satellite observation for thin aerosol layer. The total number of validation points (14711, 14031, and 521) from this algorithm (Fig. 6 (b), (e), and (h)) is about 6-7% higher than those (13714, 13137, and 493) from the original algorithm (Fig. 6 (a), (d), and (g)), which means that the new algorithm successfully retrieved the aerosol in more cases than the original algorithm. Here, the total number of validation points for $\omega$ is less than those for $\tau$ and $\alpha$, because the number of $\omega$ data from AERONET inversion products is less than those of $\tau$ and $\alpha$ from the direct sun measurements.

[P8_L225]

In addition, the retrieval results around the red circles show that the new algorithm successfully retrieved the $\sigma_{\tau_a}$, $\sigma_{\eta_{f_a}}$, $\sigma_{\eta_{c}^{dst}{}_a}$ even where the original algorithm failed to retrieve.

Further, we investigate the cause of the possible large deviation of validation results of this algorithm, and added Fig.8-11 in the revised manuscript.

[P8_L247]

We also investigated the cause of the possible large deviation between the retrieved parameters from the new algorithm and the ground observation. Figures 8, 9, and 10 show the validation results of $\tau$, $\alpha$, and $\omega$, respectively, when the chi-square value ($\chi^2$) and the uncertainties of the retrieved three parameters ($\tau$, $\eta_f$, and $\eta_c^{dst}$) are smaller than a threshold. The chi-square value ($\chi^2$) is calculated as follows:

$$\chi^2[R - F(x)] = [R - F(x)]^T S_e^{-1}[R - F(x)]/N, \quad (7)$$

and shows the closeness of the retrieved value to the observed value. The covariance matrix of the uncertainties of the retrieved parameters $S_{\hat{x}}$ is calculated using the law of error propagation, as follows:

$$S_{\hat{x}} = (A^T S_e^{-1} A)^{-1} \; , (8)$$

where $A$ is the Jacobian matrix. $S_e$ is the covariance matrix of $R$, and calculated from sum of sensor noise and the uncertainty in TOA reflectance that results from surface reflectance uncertainty (Yoshida et al., 2018). In reality, the $S_e$ is almost determined by the uncertainty in TOA reflectance that results from surface reflectance uncertainty, because sensor noise is much smaller. Therefore, the $S_{\hat{x}}$ is mostly caused by the surface reflectance uncertainty. Figure 8 shows that RMSE for $\tau$ decreases as the threshold of $\chi^2$ or $S_{\hat{x}}$ becomes strict (i.e., decreases). On the other hand, RMSE for $\alpha$ (in Fig. 9) is not dependent on the threshold of $S_{\hat{x}}$, but decreases as the $\chi^2$ threshold decreases. RMSE for $\omega$ (in Fig. 10) is little dependent on the threshold of $S_{\hat{x}}$ and $\chi^2$. Next, in Fig. 11 we investigated how the retrieved accuracy (difference between aerosol parameters retrieved from AHI and those of AERONET) depends on the model's (i.e., a priori) accuracy. The retrieved accuracy of $\alpha$ and $\omega$ has strong linear relationships (a correlation of 0.801, and 0.739, respectively) to the model's accuracy, while that of $\tau$ has a moderate linear relationship (a correlation of 0.622). Summarizing these results, the retrieved accuracy of $\tau$ depends on all of the closeness to the observed value, accuracy of the surface reflectance estimation, and accuracy of a priori estimate, while the accuracy of a priori estimate is critical for the retrieved accuracy of $\alpha$ and $\omega$. Thus, introducing more realistic a priori estimates in the new retrieval algorithm instead of the constant values in the original algorithm led to the improvement of RMSE. It is also shown that the improvement of a numerical aerosol forecast by improving the aerosol transport model and the assimilation method, and increasing the assimilation frequency may further improve the retrieval accuracy in the future.

[Figure]

**Figure 8: Frequency distribution of $\tau$ retrieved from AHI and those from AERONET. The results retrieved from this algorithm in the case of $\chi^2$ less than 20, 0.5, 0.2, and uncertainties of the retrieved $\tau$ ($S_\tau$) less than 20, 1.0, 0.5 are plotted in each panel. E, B, and R above the figures show the root mean square error, mean bias, and correlation, respectively.**

[Figure]

**Figure 9: Frequency distribution of α retrieved from AHI and those from AERONET. The results retrieved from this algorithm in the case of $\chi^2$ less than 20, 0.5, 0.2, and uncertainties of the retrieved$\eta_f$ (S$_{\eta_f}$) less than 20, 0.5, 0.2 are plotted in each panel. E, B, and R above the figures are the same as in Fig. 8.**

[Figure]

**Figure 10: Frequency distribution of $\omega$ retrieved from AHI and those from AERONET. The results retrieved from this algorithm in the case of $\chi^2$ less than 20, 0.5, 0.2, and uncertainties of the retrieved $\eta_c^{dst}$ ($S_{\eta_c^{dst}}$) less than 20, 0.5, 0.2 are plotted in each panel. E, B, and R above the figures are the same as in Fig. 8.**

[Figure]

**Figure 11: Frequency distribution of the difference between $\tau$ (a), $\alpha$ (b), and $\omega$ (c) retrieved from AHI and those from AERONET, as a function of the difference between $\tau$ (a), $\alpha$ (b), and $\omega$ (c) of a priori estimate and AERONET. R shows the correlation.**

**Other corrections**

· **English correction**

The revised manuscript was corrected by native speakers once again, and shown in green in the revised manuscript.

· **Adding explanation**

We added 'Satellite' to the title in order to express the manuscript's content accurately.

The revised title is as follows:

Satellite Retrieval of Aerosol Combined with Assimilated Forecast

We added the following sentence to avoid misunderstanding.

[P4_L104]

The obtained retrieval at T1 is further used to derive the retrieval at the following time step (T=T2, not shown) by using L4 forecasts for a priori in the same manner.

· **Affiliation change**

We changed the first author's affiliation after the discussion between the previous and present affiliation, because this paper was the results of the previous affiliation (Japan Aerospace Exploration Agency). The first author's affiliation of the revised paper is as follows.

[1] Japan Aerospace Exploration Agency, Tsukuba, 305-8505, Japan, (Present affiliation is Remote Sensing Technology Center of Japan, Tsukuba, 305-8505, Japan)